# Foliarly Applied 24-Epibrassinolide Modulates the Electrical Conductivity of the Saturated Rhizospheric Soil Extracts of Soybean under Salinity Stress

**DOI:** 10.3390/plants11182330

**Published:** 2022-09-06

**Authors:** Victoria Otie, Ali Ibrahim, Itohowo Udo, Junichi Kashiwagi, Asana Matsuura, Yang Shao, Michael Itam, Ping An, Anthony Egrinya Eneji

**Affiliations:** 1Department of Soil Science, Faculty of Agriculture, Forestry and Wildlife Resources Management, University of Calabar, P.M.B. 1115, Calabar 540271, Nigeria; 2Arid Land Research Centre, Tottori University, Hamasaka, Tottori 6800001, Japan; 3Research Faculty of Agriculture, Hokkaido University, Kitaku, Kita9 Nishi9, Sapporo 0608589, Japan; 4Faculty of Agriculture, Shinshu University, 8304, Minamiminowa-Village, Kamiina-County, Nagano 3994598, Japan; 5College of Plant Science & Technology, Huazhong Agricultural University, Wuhan 430070, China; 6Department of Plant, Soil and Microbial Sciences, Michigan State University, East Lansing, MI 48824, USA

**Keywords:** arid region, soybean, 24-epibrassinolide, root exudates, rhizosphere, salinity stress

## Abstract

The accumulation of salts within the rhizosphere is a common phenomenon in arid and semi-arid regions where irrigation water is high in salts. A previous study established the ameliorative effect of foliarly applied 24-epibrassinolide (BR) on soybean under salinity stress. As a follow-up to that study, this work evaluated the effects of BR on the electrical conductivity of saturated soil extracts (EC_se_s) under soybean exposed to salt stress. Three salinity levels (3.24, 6.06 and 8.63 dS/m) in a factorial combination with six frequencies of BR application—control, seedling, flowering, podding, seedling + flowering and seedling + flowering + podding—were the treatments, and the rhizospheric EC_se_ was monitored from 3 to 10 weeks after the commencement of irrigation with saline water (WAST). The principal component analysis revealed that samples in saline BR treatments clustered together based on the BR application frequencies. There was a significant increase in EC_se_ with increases in salinity and WAST. The frequent application of BR significantly reduced EC_se_ to 5.07 and 4.83 dS/m relative to the control with 6.91 dS/m, respectively, at week 10. At 8.63 dS/m, the application of BR (seedling + flowering + podding) reduced EC_se_ by 31.96% compared with the control. The underlining mechanism is a subject for further investigation.

## 1. Introduction

Soil salinization is one of the major abiotic stresses encountered in agricultural environments. Its management can be achieved through the leaching and planting of salt-tolerant crops. Crop production in saline soils is inhibited by specific ions, as well as osmotic and toxic effects. Managing saline soil requires sustainable practices that maintain and improve the physicochemical properties of the soil, while attaining optimum production [1]. The major source of water for crop production in drylands is rainfall, which is often limited and irregular, and results in short-term crop productivity rather than long-term soil resource sustainability [2]. In semi-arid and arid regions, it is important to continuously monitor the electrical conductivity of the saturated soil extract (EC_se_) in the rhizosphere to aid in the management of soil salinization for better crop production. It was previously reported that some drylands had high salt concentrations that rendered them inappropriate for agricultural purposes [3].

It was speculated that the world’s population may increase from 6 to approximately 10 billion by 2050, and to meet the food demand of this growing populace, a larger increase in food production is inevitable [4]. The advantages of amending sandy saline soils with organic inputs are generally short-lived due to its rapid decomposition under intense temperature and aeration, thereby causing a frequent yearly application to maintain soil productivity. Furthermore, improving sandy saline soils with inorganic fertilizers could often be unaffordable to peasant farmers [1,5]. A more stable compound such as 24-epibrassinolide (BR) could be used as an alternative to mitigate the deleterious effects of salinity on plant growth. Although considered a plant hormone that could regulate an array of physiological and developmental processes in plants under abiotic soil stressors [6,7,8,9,10], it was also found to modulate the biosynthesis of plants’ primary metabolites, such as photosynthetic pigments, monosaccharides and protein, which are secreted as exudates through plant roots to the rhizosphere [11]. Exudates secreted from plant organs, including the roots, are involved in diverse interactions within the rhizospheric soil, such as chelating toxic compounds from the soil and changing the soil pH to help plants cope under stressed soil conditions [12].

In our previous report [10], we showed that BR caused significant increases in dry matter accumulation, protein content, photosynthetic activities, nutrient uptake and partitioning, as well as other physiological attributes and seed yield of soybean under salinity stress. However, there is a paucity of information on the influence of BR-treated plants on the rhizospheric electrical conductivity of the saturated soil extract (EC_se_). Here, we hypothesized that BR could mediate the composition of root exudates to the advantage of plants growing in such stressed soils. This study aimed at evaluating the effects of the foliar application of BR at different growth stages of soybean, under varying salt concentrations on the rhizosphere EC_se_.

## 2. Materials and Methods

### 2.1. Site Trial Management

This work is part of a previously published work [10]. Briefly, the experiment was conducted in a glasshouse of a subdivision of plant eco-physiology at the Arid Land Research Center (ALRC), Tottori University, Japan. The top soil (0–15 cm) used was analyzed in the previous study and the results are presented in Section 3.1 of this paper (Table 1). The treatments studied were three levels of salinity (3.24, 6.06 and 8.63 dS/m) in combination with six frequencies of 24-epibrassinolide (BR) applications {no application (control) and applications at seedling (when the trifoliolate leaves unfolded), flowering (when one flower opened at any node on the main stem), podding (when pods began to emerge on one of the four uppermost nodes on the main stem), seedling + flowering and seedling + flowering + podding stages) and eight monitoring durations of EC_se_ (3, 4, 5, 6, 7, 8, 9 and 10 weeks after the commencement of irrigation with saline water (WAST), which was the gestation period of the test crop—12 weeks)}. Treatments were laid out as a 3 × 6 × 8 factorial, arranged in a completely randomized design (CRD) and replicated five times.

Soybean (cv. Tachiyutaka8428h) seeds were obtained from Denki nojo, Yamagata, Japan, while the BR was obtained from XenanXinyu Chemical Technology Company Limited (Ltd), Henan Province, China. The NaCl was procured from Gourmet Meat World Company Ltd., Tottori city, Japan. The three salinity levels (SLs = 3.24, 6.06 and 8.63 dS/m) were obtained by dissolving 1.75, 3.51 and 5.26 g of salt (NaCl), respectively, in 1 liter (L) of distilled water, and the corresponding concentration read from a hand-held CM-20E, TOA electrical conductivity meter (Tokyo TOA Electronics Limited (Ltd), Japan). Calibrated plastic pots with a height of 30 cm and diameter 25 cm, perforated at the bottom, were filled with 20 kg of sanddune soil and labelled according to treatment combinations. Soybean seeds were surface sterilized in 5 % sodium hypochlorite solution for 10 min and rinsed thoroughly in three changes of distilled water before sowing. Three seeds were sown per pot and later thinned to one vigorous plant at 10 days after sowing (DAS). Liquid fertilizer solution (NPK HYPONEX (6:10:5) produced by Kabushiki Corporation, Tokyo, Japan, was diluted at the rate of 5 mL in 1000 mL distilled water and applied as a blanket treatment on all the plants at 200 mL per pot, 2 and 5 weeks after sowing (WAS).

The saline treatment started at 3 WAS by irrigating plants with saline water (500 mL/pot) based on the treatment specification each day until the end of the experiment. The BR solution was prepared by dissolving 1 g of 24-epibrassinolide powder in 20 L of distilled water and foliar sprayed at the rate of 200 mL per plant according to the manufacturer’s recommendation. The application of BR was performed at 20 days after sowing (DAS) (seedling stage), 30 DAS (flowering stage) and 40 DAS (podding stage) using a calibrated hand sprayer.

### 2.2. Data Collection

The electrical conductivity of the saturated rhizospheric soil extract (EC_se_) was determined starting from the first day of salinity treatment, and thereafter on a weekly basis till the tenth week after salinity treatment with the use of an electrical conductivity meter (CM-20E, TOA, Tokyo TOA Electronics Ltd., Kobe, Japan).

### 2.3. Statistical Analysis

To assess the interrelated effects of BR, SL and WAST on the electrical conductivity of the saturated soil extracts, a principal component analysis (PCA) was conducted using the FactoMineR package in R [13]. The data were further subjected to an analysis of variance using GenStat software edition 15.1 to partition the effects of the three factors and their interactions. Treatment means were compared using Duncan’s new multiple range test at the 5% level of probability.

## 3. Results

### 3.1. Soil Analysis

Table 1 presents the results of the analysis of the physicochemical properties of the experimental soil. The soil was dominated by sand, followed by silt and clay. It was classified as *Entisols*, which are weather-resistant and can withstand extreme wetness or dryness. The soil was moderately acidic in the reaction, low in organic carbon, available phosphorus, total nitrogen, exchangeable cations (NH_4_^+^, K^+^, Ca^2+^ and Mg^2+^) and electrical conductivity of the saturated soil extract (EC_se_). The soil also had a low (2.00 meq/100 g) cation exchange capacity, indicating poor inherent fertility.

### 3.2. Principal Component Analysis

The principal component analysis (PCA) revealed differences in the electrical conductivity of the saturated soil extracts (EC_se_) obtained from different treatments. The principal components one (Dim 1) and two (Dim 2) explained 72.8% and 10.0%, respectively, of the variation in the EC_se_. Dim 1 separated soil extracts based on salinity levels (SLs), while Dim 2 separated them based on BR application frequencies (Figure 1a). The soil extracts without salt stress (control = SL_0_) clustered together irrespective of BR application, whereas the salt-stressed (SL_n_) samples clustered together based on the BR application frequencies, indicating the effectiveness of BR on the salt-treated extracts. Most of the treatment combinations had a similar contribution to the overall variation in the soil samples (Figure 1a). The soil extracts were also separated based on weeks after the commencement of irrigation with saline water (WAST) (Figure 1b). The samples from weeks 3 to 10 were separated on Dim 1. Most replicates from the same time point clustered together on the PCA, indicating sampling accuracy.

#### 3.2.1. Effects of Salinity Levels, 24-Epibrassinolide and Weeks after the Commencement of Irrigation with Saline Water (WAST) on the Electrical Conductivity of the Saturated Soil Extracts in the Rhizosphere (EC_se_)

The main effects of salinity levels (SLs), 24-epibrassinolide (BR) and weeks after the commencement of irrigation with saline water (WAST) on the electrical conductivity of the saturated soil extracts in the rhizosphere (EC_se_) are shown in Table 2. An increase in the salt concentration caused a significant (*p* ≤ 0.05) increase in EC_se_. The application of BR, irrespective of the growth stage, significantly (*p* ≤ 0.05) reduced EC_se_ relative to no application (BR_0_). The application of BR at BR_1_, BR_2_, BR_3_, BR_4_ and BR_5_ reduced EC_se_ by 10.04%, 10.66%, 11.07%, 18.03% and 21.72%, respectively, compared with the control. However, frequent BR applications (BR_5_—application at seedling + flowering + podding stages) caused the most significantly lowest salt concentrations in the rhizosphere. A successive increase in the weeks after irrigation with saline water led to a corresponding increase (*p* ≤ 0.05) in the EC_se_ of the rhizospheric soil.

#### 3.2.2. Interaction Effects of Growth Stages of Application of BR and Salinity Concentrations on Electrical Conductivity of the Saturated Soil Extracts (EC_se_s) in Rhizospheric Soil

Table 3 shows the interaction effects of the growth stages of the application of BR and salinity concentrations on the electrical conductivity of the saturated soil extract (EC_se_s) in the rhizospheric soil of soybean. At each stage of BR application, including the BR_0_ (control), increases in salinity levels (SLs) significantly increased the EC_se_. At all SLs, the BR application reduced (*p* ≤ 0.05) the rhizospheric EC_se_ relative to no application (BR_0_). However, at each SL, a one-time application of BR (i.e., BR_1_—seedling stage; BR_2_—flowering stage; BR_3_—podding stage) had a statistically similar rhizospheric EC_se_. The application of BR at BR_4_, the seedling + flowering stage, or BR_5_, reduced (*p* ≤ 0.05) the rhizospheric soil EC_se_ compared with other stages of application. At the highest concentration of salinity (8.63 dS/m), the application of BR at BR_1_, BR_2_ BR_3_, BR_4_ and BR_5_ reduced EC_se_ by 12.46%, 11.95%, 12.46%, 19.87% and 23.23%, respectively, relative to the control.

#### 3.2.3. Interaction Effects of Growth Stages of BR Application and Weeks after Commencement of Irrigation with Saline Water (WAST) on EC_se_

The interaction effects of the growth stages of BR application and weeks after the commencement of irrigation with saline water (WAST) on EC_se_ are shown in Table 4. There was a significant increase in rhizospheric EC_se_ with successive increases in WAST with no BR applied (BR_0_) and when applied only once at BR_1_, BR_2_ and BR_3_. However, when BR was applied twice (BR_4_) or thrice (BR_5_), the rhizospheric EC_se_ was statistically similar from week 7 to week 10. At weeks 4, 5 and 6 after the commencement of saline water irrigation, the BR_1_, BR_2_, BR_3_ and BR_4_ treatments had statistically similar rhizospheric EC_se_s. However, from week 7 to week 10, BR_4_ and BR_5_ significantly reduced the rhizospheric EC_se_ relative to other stages of application. At week 10, the BR application at BR_1_, BR_2_, BR_3_, BR_4_ and BR_5_ reduced EC_se_ by 16.21%, 16.93%, 16.93%, 26.63% and 23.23%, respectively, relative to no application.

#### 3.2.4. Interaction Effects of Salinity Levels and Weeks after Commencement of Irrigation with Saline Water

Table 5 shows the interaction effects of salinity levels (SLs) and weeks after the commencement of irrigation with saline water on the electrical conductivity of the saturated soil extracts. For each salinity concentration, successive increases in weeks after saline water irrigation led to corresponding significant increases in rhizospheric EC_se_. The same trend was observed at each week with successive increases in salinity.

#### 3.2.5. Three-Way Interaction Effects of BR Application Stages, Salinity Levels and Weeks after Commencement of Irrigation with Saline Water on the Electrical Conductivity of the Rhizospheric Saturated Soil Extracts

As Table 6 shows, although the interactions among the factors were not significant (*p* > 0.05), with or without BR application, there was an increase in EC_se_ from week 3 to week 10, irrespective of salinity level. The highest EC_se_ was observed in the rhizosphere with no BR application and irrigated with saline water (SL_3_ = 8.63 dS/m) at the tenth week (week 10). There was a great decrease in EC_se_ with frequent applications of BR (BR_4_ and BR_5_) at each respective salinity level and weeks after saline water irrigation, relative to other growth stages of BR application. At week 10 after the commencement of irrigation with saline water, and at the highest salinity concentration of 8.63 dS/m, BR application at BR_1_, BR_2_, BR_3_, BR_4_ and BR_5_ reduced EC_se_ by 20.53%, 20.78%, 21.14%, 30.01% and 31.96%, respectively, compared with the control.

## 4. Discussion

The monitoring of the in situ electrical conductivity of saturated soil extracts (EC_se_s) of rhizospheric soils is an important management tool aimed at determining the extent and severity of salinization processes. Overtime, the extent of the salinization of croplands may increase, resulting in accelerated land degradation and desertification, decreased farm output and, consequently, jeopardizing the environment and food security, particularly in drylands [14]. Such soils lack distinct horizons, an optimal condition for agricultural soils and are generally low in organic matter/carbon, natural fertility and water-holding capacity [15].

Increases in salinity levels significantly elevated the EC_se_ in the soil rhizosphere of soybean plants and peaked at the highest concentration. This was possibly a result of excessive accumulations of Na^+^ and Cl^−^ due to high water evaporation; such a concentration of salts could possibly induce ion toxicity [16] in soils. Excessive Na^+^ in the soil would decrease the uptake of NH_4_^+^ and other base cations (K^+^, Ca^2+^ and Mg^2+^) from the soil’s exchange complex. Although, Cl^−^ is not readily adsorbed, it does move with soil–water and is easily taken up by plants through enzymes in the root cell membrane, thereby reducing the absorption of nitrates from the soil [17]. It was estimated that by the year 2050, more than 50% of the arable land across the globe would have been salinized [18]. However, various agronomic practices, including exogenously applied growth regulators, can mitigate the adverse effects of different abiotic stresses such as drought, heavy metals as well as salt stress [10,19,20].

24-epibrassnolide (BR), an active by-product from brassinolide biosynthesis is one of the stress ameliorative approaches adopted to improve and stimulate different plant metabolic processes. It is known to improve the efficiency of transpiration and accelerates metabolic processes that favor the accumulation and release of carbohydrates in the form of gluconic acids, amino acids and organic acids [21] in plant organs (leaves, shoots or roots) as exudates to their surrounding media. These compounds are involved in numerous interactions within the rhizosphere, and could contribute to the decline in EC_se_ and salts, including Na^+^ and Cl^−^ [22].

In this study, the separation of salt extract on the PCA based on BR applications demonstrated its effectiveness in reducing the EC_se_ of saline soils. It further proved to be sufficient in ameliorating the detrimental effects of salinity on soybean when applied at maximum frequencies of BR_4_ and/or BR_5_ growth stages. The sampling accuracy indicated by the clustering of similar data points on the PCA further confirmed these findings. The effects could have occurred through changes in salt concentrations in the rhizospheric zones due to exudates from plant roots. Exudates are primary and secondary metabolites (carbohydrates, amino acids, organic acids, flavonoids, glucosinolates, auxins, etc.) secreted by different plant organs, including roots, to their surrounding environment [23]. These metabolites have been identified and quantified in different plant species, including soybean or the common bean [24,25]. The exogenous application of BR was recently reported to boost the yield and dry matter (DM) production in soybean and rice under salinity or high-temperature stress [10,26]. The increased DM production could enhance the secretion of exudates from plant roots to complex with salt ions and significantly reduce the EC_se_ in the rhizospheric soil. Root exudates (flavonoids) obtained from soybean and the common bean whose concentrations increased when plants were under salt stress [27,28] could mitigate both osmotic and ion toxicity in the rhizosphere. Similarly, the excised roots of almond trees cultured under in vitro conditions exuded proline in larger quantities when under salt stress [29].

There was a substantial increase in EC_se_ across the weeks after the saline water (WAST) treatment. In saline drylands, where high evaporation rates tend to concentrate the water solution, natural salinization can occur regardless of anthropogenic activities, especially when cations and anions are easily leached down the aquifers, resulting in a relative increase in Na^+^ ions, which, consequently, may replace Ca^2+^ and Mg^2+^ in the soil exchange complex [30]. Although plants take up many salts in the form of nutrients, when more salt is added to the soil than required, it poses a threat to plants. For the various salt concentrations used in this study, there was a gradual increase in EC_se_ with the number of weeks after the commencement of the salinity treatment, becoming extremely high at the peak of the duration (week 10). Such a high EC_se_ (as a result of excessive sodium ions accumulated overtime) may reduce the soil’s ability to conduct water and cause the accumulation of a salt crust on the soil’s surface that could leach down during an unexpected precipitation to cause injury to plant roots [31].

The interaction between BR and salinity levels showed that an increase in BR application frequency significantly reduced the EC_se_ across various salinity levels. The exogenous application of BR was reported to reduce Na^+^ and Cl^−^ contents, while increasing N, P, K and Ca to boost the activities of antioxidant enzymes, leading to improved plant yield under salinity stress [32]. The capacity of BR to play this mitigating role could possibly be attributed to its ability to induce the secretion of metabolites from different parts of the plant root system into the soil environment during growth [33]. In a recent study, the contents of primary metabolites, such as photosynthetic pigments, monosaccharides and protein, improved the growth rate of duckweed (*Wolffia arrhiza*) when sprayed with BR [34]. Root exudates serve as a material exchange and information transmission between plants and the soil in maintaining the functionality and vitality of the rhizosphere [35]. The root secretions could alter the rhizospheric soil environment, which is a major adaptative response mechanism for plants to cope with environmental abiotic stresses such as salinity [36].

The temporal effects of 24-epibrassinolide on the EC_se_ with the number of weeks after the commencement of irrigation with saline water confirmed the efficacy of BR in accelerating metabolic processes that enhanced the accumulation of soluble proteins, and the stability or synthesis of organic acids and monosaccharides, mainly glucose, during photosynthesis [37]. This was evident across the weeks of saline water irrigation, where it significantly reduced the EC_se_ to boost soybean yield across the weeks of measurement. The higher frequency of BR application may have promoted the synthesis of metabolites, thereby contributing to an increased plant biomass and, possibly, root exudates, as beneficial support to soybean growth under prolonged salinity stress.

Soils are complex entities that provide substrates for nature as well as agriculture. Saline soils inhibit the growth of most crops because of the higher concentration of neutral soluble salts in them. This occurs majorly through irrigation water, which is usually a gradual process where the salts accumulate over time before their effects are visible. Most salts are taken up by plants in the form of nutrients, but when the soil receives more salt than is removed, plants are eventually affected by salinity injuries [38]. The increase in salt concentration in our rhizospheric soil increased the EC_se_ as the saline irrigation week progressed. The plants showed some growth inhibitions from the SL_2_ (6.06 dS/m) at week 7, whereas a greater growth inhibition occurred at week 5 when the salt concentration was highest at SL_3_ (8.63 dS/m) as compared with SL_1_ (3.24 dS/m) [10]. This trend could be attributed to the fact that the soil physicochemical properties and ecological balance were altered by excessive salt accumulations overtime, leading to high osmotic stress, nutritional disorders and toxicities in the plants [39]. Soybean is considered to have a moderately tolerant threshold of approximately 5.0 dS/m of saturated soil extract [40]. The highest frequency of BR application (BR_5_) was able to reduce EC_se_ to 4.99 dS/m at week 9 and 4.93 dS/m at week 6 with saline irrigation water of 6.06 and 8.63 dS/m, respectively. This indicated that the plant was substantially protected at its vegetative and reproductive growth stages when considering the 5.0 dS/m EC_se_ tolerance threshold.

There was an improvement in the general morphological and physiological wellbeing of soybean plants when BR was exogenously applied. As reported previously [41], the continuous application of BR improved the biosynthesis of targeted compounds in the plants and facilitated the secretion of primary metabolites that suppressed the abiotic environment. The successive increase in salinity increased the EC_se_ across the weeks of saline treatment. The highest EC_se_ was under SL_3_, without BR. The application of BR reduced the EC_se_ substantially between weeks 5 and 7 at SL_2_ or SL_3_ as compared with BR_0_. However, repeat use of BR (BR_4_ or BR_5_) significantly reduced the EC_se_ up to week 9, whether at SL_2_ or SL_3_, relative to BR_1_, BR_2_ or BR_3_. As previously reported [10], soybean responds to different salinity levels in terms of both the total biomass yield and yield variables. Generally, a lower yield was observed at higher salinity levels. The BR aided the mobilization of nutrients through the exuded metabolites. Plants add glucose to rhizospheres through exudation to reduce electrolyte concentrations and facilitate their survival in saline drylands [42].

## 5. Conclusions

Our study concluded that 24-epibrassinolide, a phytohormone that regulates several physiological processes and increases plant contents of primary metabolite, could also confer tolerance to salinity stress by modulating the electrical conductivity of saturated soil extract (EC_se_) in the rhizosphere of soybean plants, possibly through root exudates. The frequent application of BR (seedling + flowering + podding) may not only mitigate the injurious effects of salt on plants, but could also reduce the high EC_se_ in the rhizosphere due to saline water irrigation. Based on the findings of this work, the quality and quantity of root exudates of BR-treated plants exposed to salinity should be evaluated to further explain its role in modulating the EC_se_.

## Figures and Tables

**Figure 1 plants-11-02330-f001:**
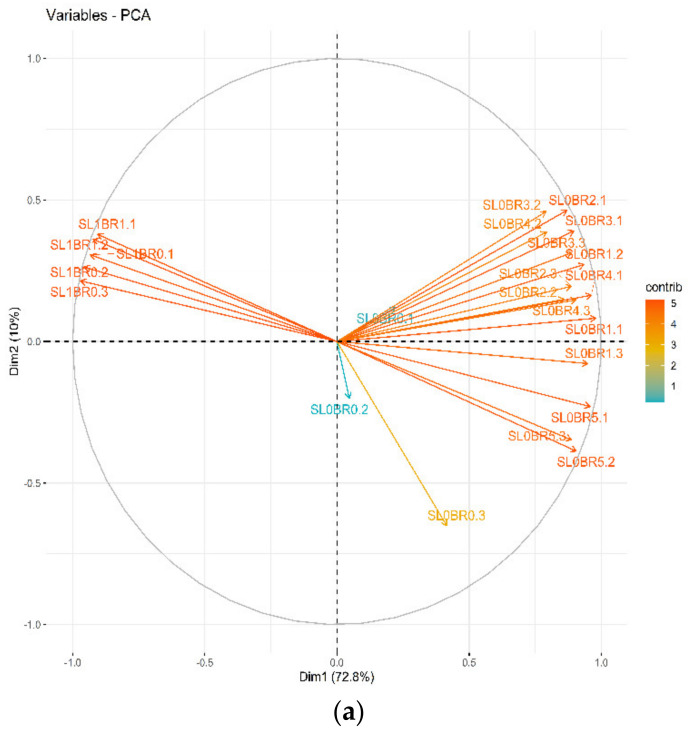
(**a**) Principal component analysis of the effects of salinity levels and 24-epibrassinolide on EC_se_; (**b**) principal component analysis on separation of soil extracts based on weeks after commencement of irrigation with saline water (WAST). SL_0_—0.11 dS/m; SL_1_—3.24 dS/m; SL_2_—6.06 dS/m; SL_3_—8.63 dS/m; BR—growth stages of 24-epibrassinolide application; BR_0_—control (no BR application); BR_1_—BR application seedling stage; BR_2_—BR application at flowering stage; BR_3_—BR application at podding stage; BR_4_—BR application at seedling + flowering stages; BR_5_—BR application at seedling + flowering + podding stages; EC_se_—electrical conductivity of saturated soil extracts.

**Table 1 plants-11-02330-t001:** Physicochemical properties of sanddune soils.

Soil Properties	Values
Sand (g/kg)	961.00
Silt (g/kg)	40.00
Clay (g/kg)	35.00
CEC (meq/100 g)	2.00
pH (H_2_O)	5.98
EC_se_ (dS/m)	0.11
C (g/kg)	0.30
P (g/kg)	0.65
TN (g/kg)	0.10
Exchangeable Cation Content (meq/100 g)	
NH_4_^+^	0.00
K^+^	0.06
Ca^2+^	0.35
Mg^2+^	0.68

TN—total nitrogen; EC_se_—electrical conductivity of saturated soil extract.

**Table 2 plants-11-02330-t002:** Main effects of salinity levels (SLs), growth stages of application of 24-epibrassinolide (BR) and weeks after commencement of irrigation with saline water (WAST) on electrical conductivity of saturated soil extracts (EC_se_s, dS/m) in soybean rhizosphere.

SL (dS/m)	EC_se_ (dS/m)	BR Application Stages	EC_se_ (dS/m)	WAST	EC_se_ (dS/m)
3.24	3.37_c_	No Application	4.88_a_	3	1.91_h_
6.06	4.38_b_	Seedling	4.39_b_	4	2.57_g_
8.63	5.15_a_	Flowering	4.36_b_	5	4.04_f_
		Podding	4.34_b_	6	4.61_e_
		Seedling + Flowering	4.00_c_	7	4.97_d_
		Seedling + Flowering + Podding	3.82_d_	8	5.20_c_
				9	5.42_b_
				10	5.68_a_

For each main effect, mean pairs with different letters were significantly different at the 5% probability level according to Duncan’s new multiple range test.

**Table 3 plants-11-02330-t003:** Interaction effects of growth stages of 24-epibrassinolide application (BR) and salinity levels (SLs) on electrical conductivity of saturated soil extracts (EC_se_s, dS/m) in soybean rhizosphere.

BR Application Stages	Salinity Levels (SLs)		
	SL_1_ (dS/m)	SL_2_ (dS/m)	SL_3_ (dS/m)
No Application	3.80_i_	4.91_c_	5.94_a_
Seedling	3.49_j_	4.49_ef_	5.20_b_
Flowering	3.47_j_	4.39_fg_	5.23_b_
Podding	3.48_j_	4.34_g_	5.20_b_
Seedling + Flowering	3.12_k_	4.13_h_	4.76_d_
Seedling + Flowering + Podding	2.89_l_	4.01_h_	4.56_e_

Mean pairs with different letters were significantly different at the 5% probability level according to Duncan’s new multiple range test. SL_1_—3.24 dS/m; SL_2_—6.06 dS/m; SL_3_—8.63 dS/m.

**Table 4 plants-11-02330-t004:** Interaction effects of growth stages of 24-epibrassinolide application (BR) and weeks after commencement of irrigation with saline water (WAST) on electrical conductivity of the saturated soil extracts (EC_se_s) in soybean rhizosphere.

BR Application Stages	Weeks after Commencement of Irrigation with Saline Water							
	3	4	5	6	7	8	9	10
No Application	2.62_r_	3.11_q_	4.47_lm_	4.95_fghi_	5.30_e_	5.63_cd_	6.07_b_	6.91_a_
Seedling	1.88_t_	2.61_r_	4.14_no_	4.70_j-l_	5.01_fgh_	5.41_de_	5.58_cd_	5.79_c_
Flowering	2.02_t_	2.53_rs_	4.04_no_	4.65_j-l_	5.05_fg_	5.32_e_	5.56_cd_	5.74_c_
Podding	1.86_t_	2.50_rs_	4.01_o_	4.64_j-l_	5.08_f_	5.31_e_	5.56_cd_	5.74_c_
Seedling + Flowering	1.56_u_	2.37_s_	3.94_o_	4.49_lm_	4.77_h-k_	4.86_f-j_	4.96_fghi_	5.07_fg_
Seedling + Flowering + Podding	1.50_u_	2.30_s_	3.64_p_	4.26_mn_	4.58_kl_	4.65_j-l_	4.76_i-k_	4.83_g-j_

Mean pairs with different letters are significantly different at the 5% probability level according to Duncan’s new multiple range test.

**Table 5 plants-11-02330-t005:** Interaction effects of salinity levels (SLs) and weeks after commencement of irrigation with saline water (WAST) on electrical conductivity of the saturated soil extracts (EC_se_s, dS/m) in soybean rhizosphere.

SL (dS/m)	Weeks after the Commencement of Irrigation with Saline Water							
	3	4	5	6	7	8	9	10
3.24	1.29_S_	1.80_q_	3.18_n_	3.67_m_	3.97_l_	4.19_k_	4.35_j_	4.55_i_
6.06	1.63_r_	2.59_p_	3.84_l_	4.87_h_	5.12_g_	5.34_f_	5.67_e_	5.96_cd_
8.63	2.81_o_	3.32_n_	5.11_g_	5.30_f_	5.81_de_	6.06_c_	6.23_b_	6.52_a_

Mean pairs with different letters were significantly different at the 5% probability level according to Duncan’s new multiple range test.

**Table 6 plants-11-02330-t006:** Interaction effects of growth stages of application of 24-epibrassinolide (BR), salinity levels (SLs) and weeks after commencement of irrigation with saline water (WAST) on electrical conductivity of the saturated soil extracts (EC_se_s) in rhizosphere of soybean.

BR Application Stages	SL (dS/m)	Weeks after Commencement of Irrigation with Saline Water							
		3	4	5	6	7	8	9	10
No application	3.24	1.83_a_	2.30_a_	3.32_a_	3.87_a_	4.28_a_	4.65_a_	4.94_a_	5.24_a_
	6.06	2.19_a_	3.03_a_	4.53_a_	5.05_a_	5.29_a_	5.75_a_	6.15_a_	7.27_a_
	8.63	3.85_a_	4.01_a_	5.56_a_	5.93_a_	6.33_a_	6.49_a_	7.11_a_	8.23_a_
Seedling	3.24	1.32_a_	1.67_a_	3.26_a_	3.79_a_	4.08_a_	4.37_a_	4.57_a_	4.85_a_
	6.06	1.71_a_	2.76_a_	3.99_a_	4.97_a_	5.11_a_	5.59_a_	5.78_a_	5.97_a_
	8.63	2.60_a_	3.40_a_	5.18_a_	5.33_a_	5.84_a_	6.29_a_	6.39_a_	6.54_a_
Flowering	3.24	1.38_a_	1.60_a_	3.21_a_	3.74_a_	4.08_a_	4.37_a_	4.55_a_	4.81_a_
	6.06	1.70_a_	2.67_a_	3.77_a_	4.94_a_	5.13_a_	5.29_a_	5.72_a_	5.89_a_
	8.63	2.97_a_	3.32_a_	5.16_a_	5.25_a_	5.93_a_	6.29_a_	6.41_a_	6.52_a_
Podding	3.24	1.30_a_	1.89_a_	3.15_a_	3.70_a_	4.04_a_	4.35_a_	4.57_a_	4.81_a_
	6.06	1.64_a_	2.41_a_	3.71_a_	4.94_a_	5.14_a_	5.25_a_	5.74_a_	5.92_a_
	8.63	2.62_a_	3.21_a_	5.18_a_	5.27_a_	6.07_a_	6.33_a_	6.38_a_	6.49_a_
Seedling + Flowering	3.24	0.99_a_	1.69_a_	3.11_a_	3.56_a_	3.83_a_	3.88_a_	3.91_a_	4.00_a_
	6.06	1.18_a_	2.38_a_	3.56_a_	4.79_a_	5.11_a_	5.19_a_	5.38_a_	5.44_a_
	8.63	2.52_a_	3.05_a_	5.16_a_	5.10_a_	5.38_a_	5.52_a_	5.58_a_	5.76_a_
Seedling + Flowering + Podding	3.24	0.91_a_	1.64_a_	3.03_a_	3.33_a_	3.49_a_	3.53_a_	3.57_a_	3.60_a_
	6.06	1.33_a_	2.30_a_	3.49_a_	4.52_a_	4.92_a_	4.99_a_	5.23_a_	5.30_a_
	8.63	2.26_a_	2.96_a_	4.41_a_	4.93_a_	5.33_a_	5.43_a_	5.49_a_	5.60_a_

Mean pairs with different letters were significantly different at the 5% probability level according to Duncan’s new multiple range test.

## Data Availability

Not applicable.

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
