# Peer review of "Foliarly Applied 24-Epibrassinolide Modulates the Electrical Conductivity of the Saturated Rhizospheric Soil Extracts of Soybean under Salinity Stress"

_plants, 2022, doi:10.3390/plants11182330_

Round 1
Reviewer 1 Report
The manuscript entitled “Can Foliarly Applied 24-Epibrassinolide Modulate the Electrical Conductivity of the Saturated Rhizospheric Soil Extracts of Soybean under Salinity Stress?'', studies that BR caused significant increases in dry matter accumulation, protein content, photosynthetic activities, nutrient uptake and partitioning, as well as other physiological attributes and seed yield of soybean, under salinity stress.
My suggestions:
The authors need to revise the title of the paper in a more meaningful way.
The abstract is written in a way lacks logic. It should highlight the salient findings more critically. Review the keywords, the keywords serve as additional indexing items to facilitate searching for the topic.
The results of this study are not fully explained therefore the interpretation of the results is very difficult. The author needs to provide the % increase or decrease rather than just writing ''significantly increased….''.
Better detail the legend of figures and tables.
The discussion is poorly written hence, needs rewriting. The discussion should be further strengthened by adding some more relevant papers. The literature search is insufficient.
The conclusion is confused. Please rewrite as it looks like results.
Reviewer 2 Report
I carefully reviewed the manuscript entitled “Can Foliarly Applied 24-Epibrassinolide Modulate the Electrical Conductivity of the Saturated Rhizospheric Soil Extracts of Soybean under Salinity Stress? [manuscript ID: plants-1878716]. And found that the manuscript topic is adequate for the aims and scope of the journal. The authors should address the following comments:
o In line 79, please mention the table instead of here.
o What were the criteria for the selection experimental durations, please mention in the 86 line.
o In line 117, the term statistics can be replaced with statistical analyses.
o It is recommended to mention the methods used for the analysis of physicochemical properties of experimental soil.
o The presentation of results is very confusing so I strongly recommend rewriting them to clearly indicate the actual meaning. Even the captions given to Tables are also so confusing at first glance, so please pay sincere attention.
o In line 288, the term anti-stress can be replaced with an appropriate one. Likewise, in 324, please replace build-up with the suitable one.
o It is suggested to add recent literature, particularly in the discussion part of the manuscript to support the findings providing the recent insights.
Round 2
Reviewer 2 Report
No further comments